# Obesity induces limited changes to systemic and local immune profiles in treatment-naive human clear cell renal cell carcinoma

**Justin T. Gibson**[1], **Katlyn E. Norris**[2], **Gal Wald**[3], **Claire M. Buchta Rosean**[3], **Lewis J. Thomas**[3], **Shannon K. Boi**[1], **Laura A. Bertrand**[3], **Megan Bing**[3], **Jennifer B. Gordetsky**[4], **Jessy Deshane**[5], **Peng Li**[6], **James A. Brown**[3,7], **Kenneth G. Nepple**[3,7], **Lyse A. Norian**[8,9,10]*

**1** Graduate Biomedical Sciences, University of Alabama at Birmingham, Birmingham, Alabama, United States of America, **2** School of Health Professions Honors Undergraduate Research Program, University of Alabama at Birmingham, Birmingham, Alabama, United States of America, **3** Department of Urology, Carver College of Medicine, The University of Iowa, Iowa City, Iowa, United States of America, **4** Departments of Pathology and Urology, Vanderbilt University Medical Center, Nashville, Tennessee, United States of America, **5** Division of Pulmonary, Allergy and Critical Care Medicine, Department of Medicine, School of Medicine, University of Alabama at Birmingham, Birmingham, Alabama, United States of America, **6** Department of Acute, Chronic, and Continuing Care, School of Nursing, University of Alabama at Birmingham, Birmingham, Alabama, United States of America, **7** Holden Comprehensive Cancer Center, The University of Iowa, Iowa City, Iowa, United States of America, **8** Department of Nutrition Sciences, University of Alabama at Birmingham, Birmingham, Alabama, United States of America, **9** O'Neal Comprehensive Cancer Center, University of Alabama at Birmingham, Birmingham, Alabama, United States of America, **10** Nutrition Obesity Research Center, University of Alabama at Birmingham, Birmingham, Alabama, United States of America

* lnorian@uab.edu

**Data Availability Statement:** All relevant data are within the manuscript and its Supporting Information files.

## Abstract

Understanding the effects of obesity on the immune profile of renal cell carcinoma (RCC) patients is critical, given the rising use of immunotherapies to treat advanced disease and recent reports of differential cancer immunotherapy outcomes with obesity. Here, we evaluated multiple immune parameters at the genetic, soluble protein, and cellular levels in peripheral blood and renal tumors from treatment-naive clear cell RCC (ccRCC) subjects ($n = 69$), to better understand the effects of host obesity (Body Mass Index "BMI" $\geq 30$ kg/m$^2$) in the absence of immunotherapy. Tumor-free donors ($n = 38$) with or without obesity were used as controls. In our ccRCC cohort, increasing BMI was associated with decreased percentages of circulating activated PD-1$^+$CD8$^+$ T cells, CD14$^+$CD16$^{neg}$ classical monocytes, and Foxp3$^+$ regulatory T cells (Tregs). Only CD14$^+$CD16$^{neg}$ classical monocytes and Tregs were reduced when obesity was examined as a categorical variable. Obesity did not alter the percentages of circulating IFNγ$^+$ CD8 T cells or IFNγ$^+$, IL-4$^+$, or IL-17A$^+$ CD4 T cells in ccRCC subjects. Of 38 plasma proteins analyzed, six (CCL3, IL-1β, IL-1RA, IL-10, IL-17, and TNFα) were upregulated specifically in ccRCC subjects with obesity versus tumor-free controls with obesity. IGFBP-1 was uniquely decreased in ccRCC subjects with obesity versus non-obese ccRCC subjects. Immunogenetic profiling of ccRCC tumors revealed that 93% of examined genes were equivalently expressed and no changes in cell type scores were found in stage-matched tumors from obesity category II/III versus normal weight (BMI $\geq 35$ kg/m$^2$ versus 18.5–24.9 kg/m$^2$, respectively) subjects. Intratumoral PLGF and

**Funding:** Work conducted within was supported by funding from the National Institutes of Health, specifically awards #T32GM811131 to JTG, #T32AI007051 to SKB, #P30DK056336 to LAN, and #5R01CA181088-06 to LAN. The funders had no role in study design, data collection and analysis, decision to publish, or preparation of the manuscript.

**Competing interests:** The authors have declared that no competing interests exist.

VEGF-A proteins were elevated in ccRCC subjects with obesity. Thus, in ccRCC patients with localized disease, obesity is not associated with widespread detrimental alterations in systemic or intratumoral immune profiles. The effects of combined obesity and immunotherapy administration on immune parameters remains to be determined.

## Introduction

Renal and pelvic cancers are among the ten most common cancers in the United States, with over 65,000 cases diagnosed in 2018 alone and approximately 23% resulting in fatality [1]. Multiple subtypes of renal cancer exist, but clear cell renal cell carcinoma (ccRCC) accounts for nearly 75% of cases [2]. In 2015, the immune checkpoint inhibitor (CPI) nivolumab, a monoclonal antibody against programmed cell death receptor-1 (PD-1), was approved for the treatment of metastatic RCC, due in part to its demonstrated ability to prolong survival relative to the targeted mTOR inhibitor everolimus [3]. In 2018, the combination of nivolumab and ipilimumab (anti-Cytotoxic T Lymphocyte Antigen-4; "anti-CTLA-4") was approved. However, objective response rates to CPI biologics remain < 50% in RCC patients [4], even when used in combination [5]. For this reason, intense efforts are underway to identify the underlying causes of suboptimal CPI efficacy.

Obesity is one of the main risk factors for ccRCC [6] and it has also been investigated as a factor that may influence both tumor progression and immune responses. Recent estimates indicate that over 39% of U.S. adults have obesity [7], defined by the World Health Organization (WHO) as a Body Mass Index (BMI) $\geq 30kg/m^2$. We and others have found that in pre-clinical models, obesity impairs protective immune responses to vaccinations and tumors and facilitates tumor progression [8–15]. Our prior studies using an orthotopic murine renal cancer model revealed that immune dysfunction was exacerbated in mice with diet-induced obesity [9, 11]. However, despite numerous pre-clinical results indicating that obesity promotes tumor progression via multiple detrimental effects on the immune system, our retrospective examination of human sarcoma subjects revealed that obesity had surprisingly limited effects on plasma cytokine and chemokine profiles, leading us to conclude that obesity, as measured by BMI, did not exacerbate the pro-tumorigenic systemic environment in treatment-naive individuals with sarcoma [16].

The relationship between obesity and CPI efficacy has also been investigated, yielding some surprising results. In 2018, two landmark studies by McQuade *et al.* and Wang *et al.* reported that obesity was associated with improved survival following CPI administration in men with melanoma [17] and men and women across mixed tumor types [18], respectively. Since that time, several papers have examined relationships between obesity and/or overweight plus obesity on CPI outcomes specifically in RCC. One study found that within the same institution, RCC patients with a BMI $\geq 25$ kg/m$^2$ (subjects with overweight or obesity) treated with CPI had a trending reduction in overall survival, whereas patients treated with therapies targeting VEGF or mTOR had improved survival [19]. In another study, CPI outcomes in RCC patients were first stratified according to primary response to therapy versus therapy resistance [20]; here, the authors found no difference in BMI between these two groups, although in the subset of patients who displayed clinical benefit, increased BMI was associated with improved progression-free survival (PFS). Finally, a 2020 report by Sanchez *et al.* found that obesity was associated with improved survival in RCC patients in response to the tyrosine kinase inhibitors pazopanib or sunitinib ($n = 256$) after adjustment for International Metastatic RCC Database

(IMDC) risk score, but not in response to CPI (*n* = 129) [21]. Thus, elucidating associations between patient obesity and responses to cancer therapy outcomes remains an important and ongoing area of investigation. Understanding the extent to which obesity alters anti-tumor immunity should contribute much-needed mechanistic insight into observed CPI outcomes in RCC patients.

It is known that variations in tumor-infiltrating leukocyte populations can dramatically alter CPI outcomes, regardless of host obesity status [22–27]. However, until the recent study by Sanchez *et al.*, it was unclear whether or how obesity influenced immune composition within renal tumors [21]. By performing a detailed transcriptomic analysis of tumor biospecimens from treatment-naive RCC subjects who either had or did not have obesity, the authors of that study determined that although obesity was associated with an increase in the angiogenesis gene expression score within renal tumors, obesity generated few changes in the intratumoral immune response. The latter finding was then confirmed by flow cytometry in a small cohort of renal tumors from RCC subjects who had undergone nephrectomy who were of normal weight (*n* = 7) or who had obesity (*n* = 16).

Here, we investigated the impact of obesity on immune profiles in treatment-naive ccRCC subjects by evaluating the cellular and soluble immune components that were present either systemically or within the renal tumor microenvironment in Stage 1–3 ccRCC subjects. We also performed nanoString immunogenetic profiling on a subset of 24 stage-matched primary ccRCC tumors and evaluated differentially expressed genes (DEG) to determine the extent to which obesity altered gene expression profiles in our study cohort. To our knowledge, our study provides the first such examination of obesity-associated changes in the systemic and intratumoral immune profiles of treatment-naive ccRCC subjects. We report that host obesity is associated with only modest changes in the multiple immune parameters we examined, illustrating that obesity as a co-morbidity does not induce widespread detrimental, immune-related alterations in treatment-naive ccRCC patients.

## Materials and methods

### Study subjects

Approval for this study was granted by the Internal Review Boards of the University of Iowa (UI) Carver College of Medicine and the University of Alabama at Birmingham (UAB). IRB review at both UI and UAB is based on the principles outlined in the Belmont Report (Ethical Principles and Guidelines for the Protection of Human Subjects of Research), written by the National Commission for the Protection of Human Subjects of Biomedical and Behavioral Research. From October 2012 to May 2015, adult patients with suspicious renal masses who were scheduled to undergo either partial or full nephrectomy for removal of renal masses were offered enrollment in an Institutional Review Board–approved study at The UI Hospitals and Clinics. Written informed consent was obtained for *n* = 162 subjects with localized renal masses (stages T1-T3) and 38 age- and BMI-matched tumor-free controls. Although subjects with all histological subtypes of RCC were consented, tumor biospecimens and peripheral blood samples were analyzed only from subjects with the ccRCC histological subtype (*n* = 69) to reduce variability. Exclusion criteria included the following: active secondary malignancy, immune-modulating medications, and metastatic disease. Demographic information including age, BMI, and clinical data (final pathology results) was obtained from the electronic health record. BMI was calculated from subject's height/weight information. WHO guidelines were used to define obesity status: non-obese (combined lean and overweight) = BMI < 30 kg/m$^2$ and obese = BMI $\geq$ 30 kg/m$^2$ (**Table 1**). For nanoString immune profiling only, we obtained and analyzed biospecimens from normal weight (BMI 18.5–24.9 kg/m$^2$) versus WHO-defined

**Table 1. Demographics information for ccRCC subjects and tumor-free donors.**

| | Renal Cancer Subjects | | Tumor-Free Donors | |
|---|---|---|---|---|
| **Obesity Status BMI (kg/m$^2$)** | **Non-Obese < 30** | **Obese ≥ 30** | **Non-Obese < 30** | **Obese ≥ 30** |
| Subjects (n) | 30 | 39 | 19 | 19 |
| Sex (%) | M 70.0% F 30.0% | M 61.54% F 38.46% | M 42.11% F 57.89% | M 47.37% F 52.63% |
| Age (years, mean) | 59.60 | 57.72 | 56.11 | 50.68 |
| Age (years, range) | 28–83 | 41–81 | 27–76 | 21–68 |
| Weight (kg, mean) | 78.53 | 121.48 | 73.26 | 118.45 |
| BMI (kg/m$^2$, mean) | 25.82 | 39.36 | 25.12 | 39.78 |
| BMI (kg/m$^2$, range) | 19.38–29.98 | 30.00–60.73 | 21.52–29.45 | 30.10–57.70 |
| Diabetes (%) | 16.67% | 38.46% | Unavailable | Unavailable |
| Stage (%) | | | | |
| T1a | 53.33% | 35.90% | | |
| T1b | 13.33% | 35.90% | | |
| T2a | 3.33% | 5.13% | | |
| T2b | 0.00% | 2.56% | | |
| T3a | 26.67% | 17.95% | | |
| T3b | 3.33% | 0.00% | | |
| N/A | 0.00% | 2.56% | | |
| Grade | | | | |
| 1 | 0.00% | 5.13% | | |
| 2 | 40.00% | 58.97% | | |
| 3 | 43.33% | 20.51% | | |
| 4 | 16.67% | 12.82% | | |
| NA | 0.00% | 2.56% | | |

obesity category II/III (BMI > 35kg/m$^2$) subjects, to remove effects of overweight and BMI category I obesity on results.

## Surface and intracellular staining for flow cytometry

Peripheral blood samples were taken in the preoperative area from all subjects. Flow cytometric analyses were performed on 30 non-obese subjects and 39 subjects with obesity who had renal tumors that were subsequently confirmed to be ccRCC, and 19 tumor-free BMI-, age-, and sex-matched controls. Peripheral blood samples were processed over Ficoll to permit mononuclear cell harvest, and frozen until use. Tumor samples obtained from surgical pathology on the day of surgery were mechanically homogenized in RPMI 1640 to a single cell suspension using a Miltenyi GentleMacs, then stained without freezing. Peripheral blood mononuclear cells (PBMCs) were thawed prior to staining and suspended in complete media (RPMI basal medium plus 10% fetal calf serum). All samples were blocked in 5% normal goat serum before antibody staining.

PBMC samples were stained with the following antibody combinations, and results were obtained using multiparameter flow cytometry on a BD LSR II (BD Biosciences) and analyzed with FlowJo software. Intracellular Foxp3 staining was performed according to the manufacturer's (BioLegend) protocol. Phenotypic leukocyte subpopulation analyses were as follows after using forward scatter vs side scatter to obtain a single cell leukocyte populations: CD4$^+$ and CD8$^+$ (BioLegend clones #OKT4 and #HIT8a respectively) PBMCs were identified as naive (CD45RO$^-$CD45RA$^+$), activated (CD45RO$^+$CD45RA$^-$), or activated PD-1 expressing (CD45RO$^+$CD45RA$^-$PD-1$^+$) (Tonbo Biosciences clone #UCHL1, BD Biosciences cline

#HI100, and BioLegend clone #EH12.2H7 respectively). We further identified non-classical monocytes (CD14$^{neg}$CD16$^+$), classical monocytes (CD14$^+$CD16$^{neg}$) (BioLegend clones #3G8 and #HCD14, respectively), and T regs (CD4$^+$Foxp3$^+$) (BioLegend clones #OKT4 and #150D respectively). For staining of fresh dissociated tumor samples the following antibodies were used to define specific populations of CD4+ and CD8+ T cells: CD3, CD4, CD8, and HLA-DR (BioLegend clones #HIT3a, #OKT4, #HIT8a, and #L243 respectively). Additionally, tumor-infiltrating myeloid populations were defined using antibodies targeting CD11b and CD33 (BioLegend clones #M1/70 and #WM153 respectively). For intracellular cytokine staining of PBMCs, frozen aliquots of cells were thawed and allowed to recover overnight in complete medium containing Benzonase. Stimulation to induce cytokine secretion was performed with Cell Stimulation Cocktail (eBioscience) for 4 hours at 37˚C. The following dyes and antibodies were used for surface and intracellular staining: Zombie Aqua (fixable live-dead), CD3, CD4, CD8, IFNγ, IL-4, and IL-17A (BioLegend clones #HIT3a, #OKT4, #HIT8a, and #4S.B3; Affymetrix clones #8D4-8 and eBIODEC17).

## Luminex analysis of soluble plasma and tumor homogenate supernatant samples

Frozen plasma samples and tumor supernatants from ccRCC subjects and plasma samples from tumor-free controls were thawed on ice and processed simultaneously to evaluate a total of 38 or 18 different soluble cytokines and chemokines, as indicated in the figures and text, according to the manufacturer's instructions (Human Inflammatory Panel and Human Cancer Panel 2 Bio-Plex assays, Bio-Rad). Tumor weight to volume ratios were kept constant at 1g tissue/10mL of RPMI 1640. Samples were run in duplicate on a Bio-Rad Bio-Plex instrument according to the manufacturer's protocol.

## nanoString immunogenetic profiling

OCT-embedded tumor blocks from 18 de-identified subjects with Stage T1 or T2 confirmed ccRCC who had category II/III obesity (BMI ≥ 35) or were of normal body weight (BMI 18.5–24.9 kg/m$^2$) were obtained from the University of Iowa Hospitals and Clinics. Only early stage ccRCC tumors were used for nanoString studies to limit variability. Six additional renal tumors from ccRCC subjects with the same disease characteristics were identified via retrospective search of the University of Alabama at Birmingham surgical pathology tissue bank. A representative paraffin embedded block of tumor was chosen from each case, cut at 5 microns, and tissue placed on unstained slides. For all tumor samples, consecutive slides were cut from each block and one slide was H&E stained then examined by a pathologist to determine presence/absence of renal tumor in the block, the percent necrotic tissue present, and to delineate tumor boundaries within the tissue block. Only specimens with > 80% non-necrotic tumor tissue were used for nanoString analysis. A total of 24 ccRCC tumor specimens from non-obese subjects ($n$ = 12) or subjects with obesity ($n$ = 12) met these criteria and were used for nanoString analysis. From these samples, identified tumor areas were harvested from all slides at the UAB nanoString facility then processed for RNA extraction and nanoString analysis using the Human PanCancer Immune Profiling kit. RNA concentration was quantitated using a Take3 micro-volume plate in conjunction with a Synergy reader (BioTek) and confirmed using a NanoDrop microvolume spectrophotometer (Thermo Fisher Scientific). Data analysis was performed using nSolver Software (nanoString Technologies Inc.) where unadjusted p values < 0.05 were used to identify differentially expressed genes.

## Statistical analyses

Statistical analyses for flow cytometry and Luminex plasma protein data were performed utilizing Prism, Version 7.00 (GraphPad). Sample sizes were determined empirically based on previously conducted studies using similar data sets and analyses [28]. Linear regression analysis was used to evaluation association between BMI and PBMC populations. Gaussian distribution of categorical data was first evaluated using the Shapiro-Wilk normality test, then analyzed using parametric one-way ANOVA test or non-parametric Kruskal-Wallis test followed by Bonferroni's or Dunn's multiple comparisons tests, respectively. When only two groups were being compared, non-parametric two-tailed Mann-Whitney U tests, parametric two-tailed unpaired Student t-tests, or parametric two-tailed paired Student t-tests were used. Prior to statistical analysis of soluble protein concentrations, statistical outliers were identified and removed using ROUT test with $Q = 0.1\%$ stringency. Outliers were not removed for any other data. Overall survival (OS) was defined as the interval from the date of nephrectomy to the date of death or to the last follow up date if patients were still alive (censored). Progression free survival (PFS) was defined as the interval from the date of nephrectomy to the date of progression or the date of death and censored at the last follow-up date if the patients were still alive without progression. Survival curves between obese and non-obese groups were compared by Kaplan-Meier analysis and logrank test. Statistical significance is denoted throughout as * $p<0.05$, **$p<0.01$, ***$p<0.001$, ****$p<0.0001$, or ns = not statistically significant. Trending non-significant p values $< 0.15$ are denoted throughout.

## Results

### Clinical characteristics of study subjects

To evaluate the impact of host obesity on peripheral and tumor immune parameters, we performed a prospective study that evaluated multiple cellular, soluble, and immunogenetic parameters in ccRCC subjects. ccRCC subjects with obesity (BMI $\geq$ 30kg/m$^2$) ($n$ = 39) had a mean BMI of 39.36 kg/m$^2$, with a range of 30–62.73 (**Table 1**). Non-obese (BMI $<$ 30kg/m$^2$) ccRCC subjects ($n$ = 30) had a mean BMI of 25.82 kg/m$^2$, with a range of 19.38–29.98. The majority of non-obese subjects (69.99%) and subjects with obesity (79.49%) had stage T1a-T2b tumors. These percentages reflect the fact that patients with disseminated disease are not typically candidates for nephrectomy and were not approached for enrollment into this study. Fuhrman grade analysis revealed that 40.00% of non-obese subjects and 64.10% of subjects with obesity had Fuhrman grade 1/2 tumors. In addition, 38 tumor-free donors enrolled in the study and were assigned to control groups of either non-obese tumor-free ($n$ = 19) or tumor-free obesity ($n$ = 19) categories, based upon BMI values as above (means of 25.12 kg/m$^2$ and 39.78 kg/m$^2$, respectively).

### Obesity is associated with decreased percentages of circulating CD14$^+$CD16$^{neg}$ classical monocytes and CD4$^+$Foxp3$^+$ regulatory T cells (Tregs)

To determine whether host obesity influenced leukocyte subpopulation composition and/or prevalence, pre-operative PBMC samples were collected for analysis by flow cytometry. We began by assessing peripheral blood leukocyte composition versus increasing adiposity, by evaluating BMI as a continuous variable in ccRCC subjects. Doing so revealed no changes in naive (CD45RO$^{neg}$CD45RA$^+$) (P = 0.829), activated (CD45RO$^+$CD45RA$^{neg}$) (P = 0.798), or activated PD-1 expressing (CD45RO$^{neg}$CD45RA$^+$PD-1$^+$) (P = 0.789) CD4$^+$ PBMCs (**Fig 1A**). Corresponding analysis of CD8$^+$ PBMCs showed no association with naive CD8$^+$ PBMCs

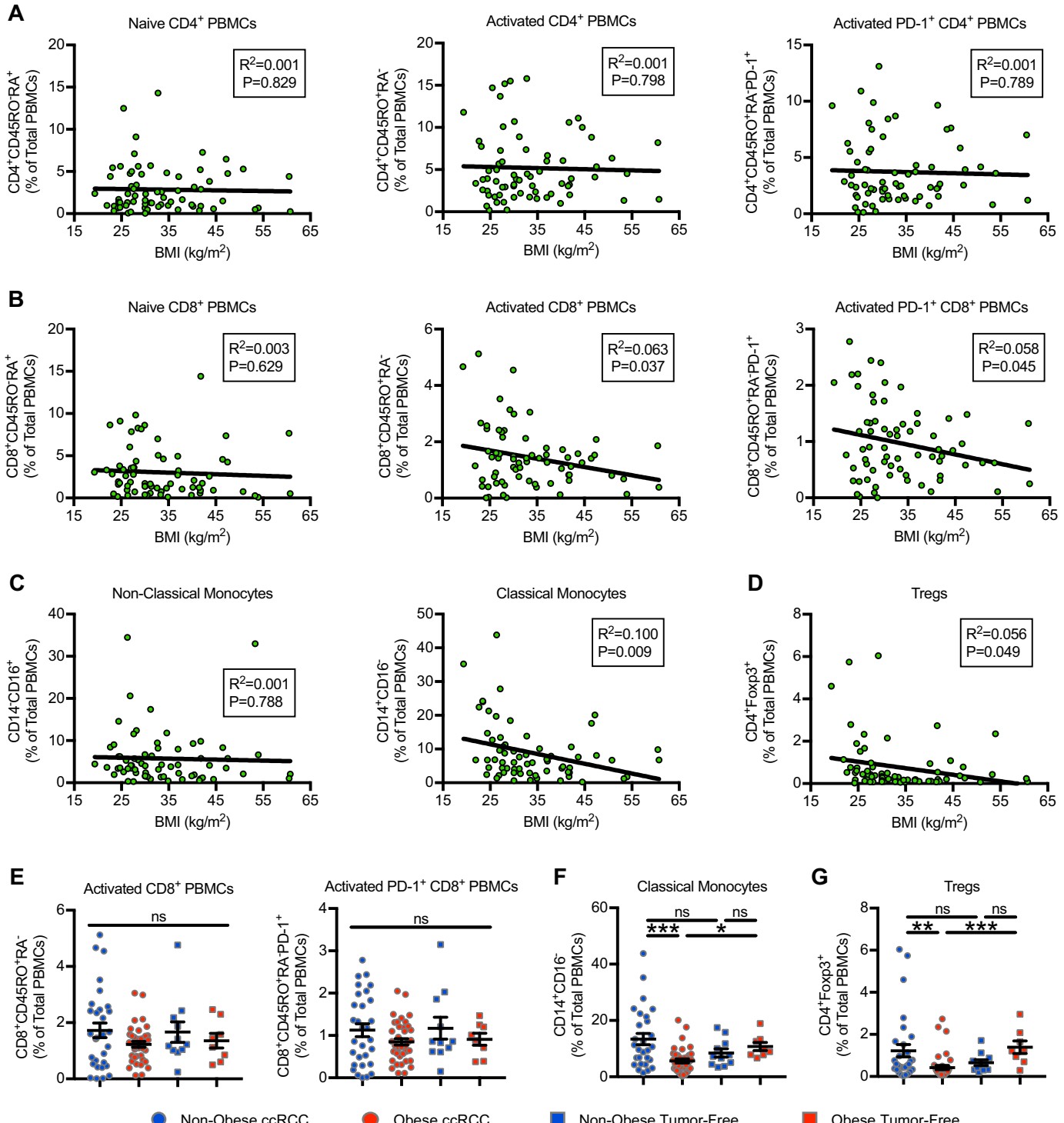

**Fig 1. Increasing BMI is associated with decreases in multiple leukocyte subsets in the peripheral blood of ccRCC subjects, whereas only classical monocytes and regulatory T cells are diminished with obesity.** Peripheral blood mononuclear cells (PBMCs) were obtained from ccRCC subjects (n = 67–69) prior to nephrectomy and analyzed via multi-parameter flow cytometry. (**A-D**) Linear regression analysis of indicated PBMC populations versus BMI for patients with ccRCC. (**E-G**) Categorical analysis of indicated PBMC populations in ccRCC subjects +/- obesity and tumor-free controls (n = 19). Data are presented as linear regression or means ± SEM with individual values for each subject shown. Statistical analyses were performed using linear regression, parametric one-way ANOVA, or non-parametric Kruskal-Wallis test followed by Dunn's multiple comparisons test. * p<0.05, ** p<0.01, *** p<0.001, ns = not statistically significant. Non-obese BMI<30 kg/m², Obese BMI≥30 kg/m².

(P = 0.629), but revealed a negative association between increasing BMI and the percentages of total activated ($R^2$ = 0.063, P = 0.037) and activated PD-1$^+$ ($R^2$ = 0.058, P = 0.045) CD8$^+$ PBMCs (Fig 1B). Evaluation of non-classical monocytes (CD14$^{neg}$CD16$^+$) showed no association with BMI (P = 0.788), whereas the percentages of classical monocytes (CD14$^+$CD16$^{neg}$) ($R^2$ = 0.100, P = 0.009) and regulatory T cell (Tregs; CD4$^+$Foxp3$^+$) ($R^2$ = 0.056, P = 0.049) PBMC populations were found to be negatively associated with BMI (Fig 1C and 1D).

We next assessed BMI as a categorical variable to determine if associations between PBMC populations of interest and WHO-defined obesity were present. Further, we also compared our ccRCC subjects to tumor-free donors, who either had or did not have obesity, to determine the impact of renal tumor presence. No associations between BMI and activated (Kruskal-Wallis, P = 0.6764) or activated PD-1$^+$ (one-way ANOVA, P = 0.2855) CD8$^+$ PBMCs from ccRCC subjects were present when obesity was examined categorically (Fig 1E). However, categorical analysis revealed a reduced percentage of classical monocytes in subjects with obesity versus non-obese ccRCC subjects (Kruskal-Wallis, P = 0.0004; Dunn's MC P = 0.0006) and tumor-free subjects with obesity (Kruskal-Wallis, P = 0.0004; Dunn's MC P = 0.0182) (Fig 1F). Similar changes were also seen in Tregs (Kruskal-Wallis, P < 0.0001; Dunn's MC P = 0.0014 and P = 0.0008, respectively) (Fig 1G). We conclude from these results that the most pronounced obesity-associated changes in systemic leukocyte composition from ccRCC subjects are reductions in the percentages of circulating classical CD14$^+$CD16$^{neg}$ monocytes and Tregs.

## Obesity does not impact cytokine secretion by CD4$^+$ and CD8$^+$ PBMCs

Given that obesity did not impact the composition of peripheral blood CD4$^+$ and CD8$^+$ populations, we next asked if it impacted their functional capacity. Bulk PBMC samples were stimulated *ex vivo* and evaluated by intracellular flow cytometry for cytokine production. Doing so revealed no differences in CD4$^+$ PBMC production of IL-4 (Kruskal-Wallis, P = 0.3402), IL-17A (Kruskal-Wallis, P = 0.9006), or IFNγ (Kruskal-Wallis, P = 0.0396; Dunn's MC P > 0.9999) in ccRCC subjects, although tumor-free donors with obesity were found to have decreased production of IFNγ compared to non-obese tumor-free donors (Kruskal-Wallis, P = 0.0396; Dunn's MC P = 0.0215) (Fig 2A). Evaluation of CD8$^+$ PMBC cytokine production revealed no changes in IFNγ production in ccRCC subjects or tumor-free donors with obesity (one-way ANOVA, P = 0.2690) (Fig 2B). These data suggest that obesity does not impact CD4$^+$ or CD8$^+$ PBMC cytokine secretion in ccRCC subjects, although mild differences were identified in tumor-free donors.

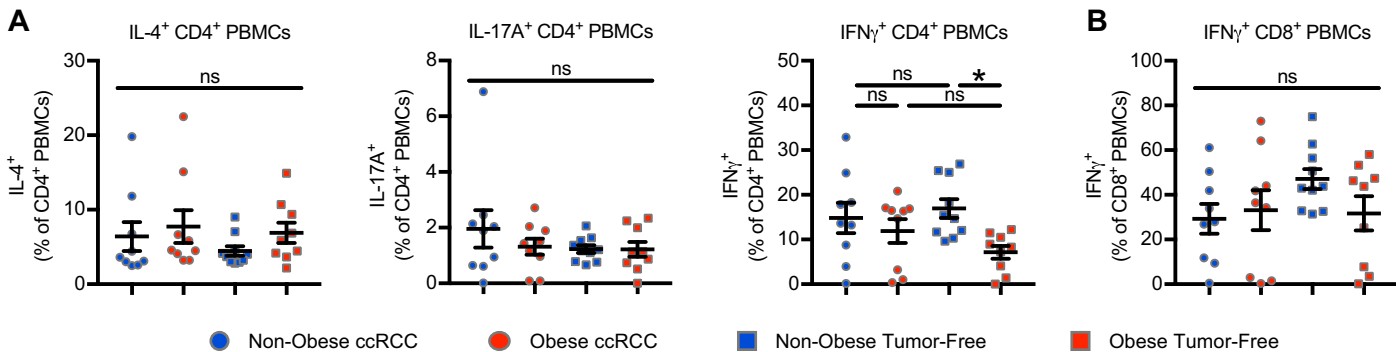

**Fig 2. Obesity minimally alters cytokine production by peripheral blood CD4$^+$ and CD8$^+$ T cells in ccRCC subjects.** PBMCs obtained from ccRCC subjects (n = 18) and tumor-free controls (n = 19) were stimulated *ex vivo* and analyzed via intracellular flow cytometry for production of indicated cytokines by (**A**) CD4$^+$ PMBCs and (**B**) CD8$^+$ PBMCs. Data are presented as means ± SEM with individual values for each subject shown. Statistical analyses were performed using parametric one-way ANOVA or non-parametric Kruskal-Wallis test followed by Dunn's multiple comparisons test. * p<0.05, ns = not statistically significant. Non-obese BMI<30, Obese BMI≥30.

## Renal tumor growth impacts systemic cytokine and chemokine profiles to a greater extent than does host obesity

We had previously determined that even early-stage renal tumors can cause systemic shifts in cytokine and chemokine concentrations [28]. Here, we asked whether host obesity skewed soluble protein profiles in ccRCC subjects. We analyzed 38 different plasma cytokines, chemokines, and growth factors known to be associated with modulation of immune function or tumor progression in a subset of ccRCC subjects (with or without obesity) and comparable tumor-free donors. Of the 38 proteins analyzed, 4 (bFGF, CCL4, CXCL10, and G-CSF) were significantly elevated ($P < 0.05$) in non-obese ccRCC subjects versus non-obese tumor-free donors, 6 (CCL3, IL-1β, IL-1RA, IL-10, IL-17, and TNFα) were significantly elevated ($P < 0.05$) in ccRCC subjects with obesity versus tumor-free donors with obesity, 12 (CCL2, GM-CSF, HB-EGF, IFNγ, IL-4, IL-5, IL-6, IL-7, IL-8, IL-12 p70, IL-13, and PDGFB) were significantly elevated ($P < 0.05$) in ccRCC subjects with or without obesity versus respective tumor-free donors, and 15 (angiopoietin, CCL5, EGF, endoglin, eotaxin, IL-18, PAI-1, PLGF, sCD40L, sFasL, TGF-α, uPA, VEGF-A, VEGF-C, and VEGF-D) were unaltered (Fig 3A). Of the 38 proteins, only one (IGFBP-1) was uniquely elevated ($P < 0.05$) in non-obese ccRCC subjects versus those with obesity, and ccRCC subjects with obesity versus non-obese tumor-free controls (Fig 3B). Interestingly, trending increases were detected for IGFBP-1 in non-obese tumor-free donors versus those with obesity ($P = 0.05$) and in non-obese ccRCC subjects versus non-obese tumor-free donors ($P = 0.08$). A single representative protein from each of the four categories in Fig 3A

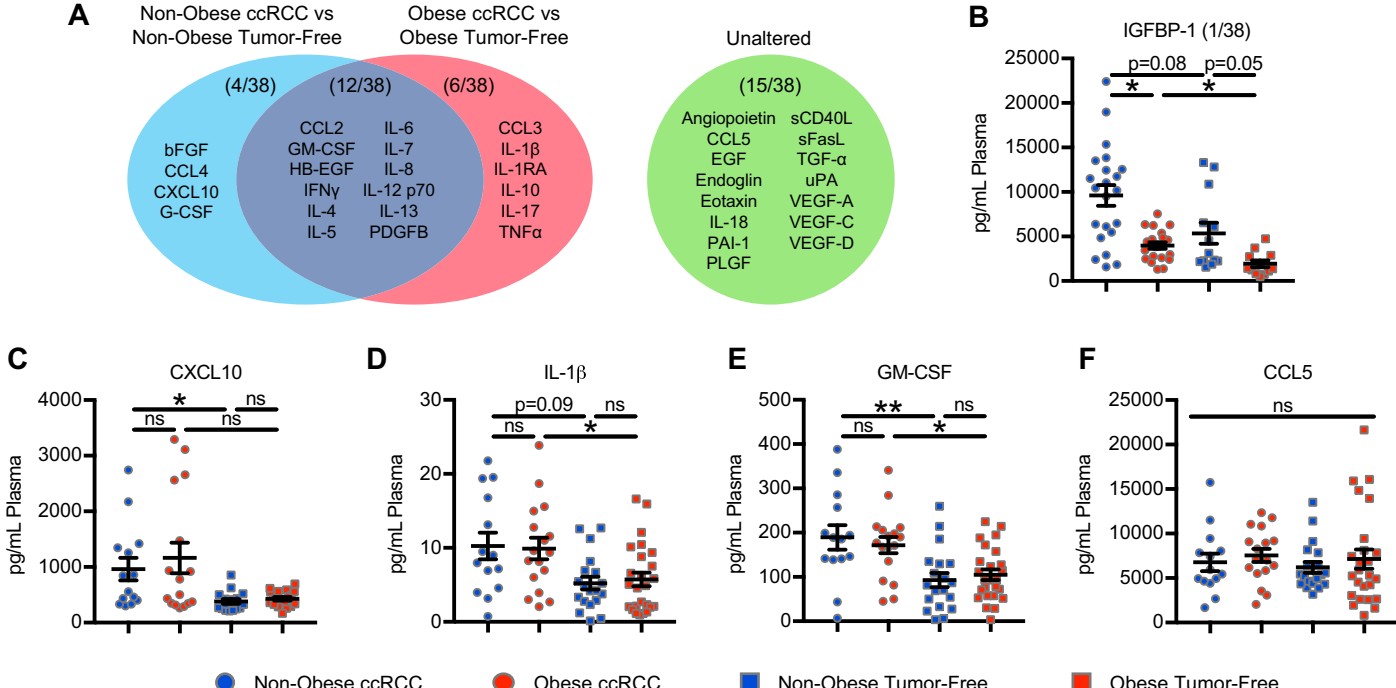

**Fig 3. Elevations in plasma proteins are primarily caused by the presence of renal tumors, rather than host obesity.** Various soluble proteins were surveyed from platelet poor plasma obtained from ccRCC (n = 15–26) and tumor-free (n = 7–16) subjects. (**A**) Of 38 proteins evaluated, 4 (blue) were elevated in non-obese ccRCC versus non-obese tumor-free subjects, 6 (red) were elevated in ccRCC subjects with obesity versus tumor-free subjects with obesity, 12 (purple) were elevated in ccRCC subjects with or without obesity versus respective tumor-free subjects, 15 (green) were unaltered, and (**B**) 1 was uniquely elevated in non-obese ccRCC versus ccRCC subjects with obesity and non-obese tumor-free controls versus controls with obesity. (**C-F**) One representative protein from each of blue, red, purple, and green groups, respectively, from panel **A**. Data are presented as means ± SEM with individual values for each subject shown. Statistical analyses were performed using parametric one-way ANOVA or non-parametric Kruskal-Wallis test followed by Bonferroni's or Dunn's multiple comparisons tests, respectively. *p<0.05, **p<0.01, ns = not statistically significant. Non-obese BMI<30, Obese BMI≥30.

(bold text) are shown (Fig 3C–3F); data for the remaining proteins that were altered can be found in S1 Fig. These data indicate that the vast majority of alterations in peripheral protein concentrations arise due to the presence of a growing renal tumor and are minimally impacted by obesity, with the exception of IGFBP-1, which was previously reported to be decreased in subjects with obesity versus those who were non-obese [29, 30] as we validate here.

## Obesity induces minor alterations to the immunogenetic profile and does not seem to alter the leukocytic composition of ccRCC tumors

Next, we investigated potential obesity-associated changes in the tumor immune microenvironment by using nanoString to perform immunogenetic profiling. To increase the likelihood of detecting obesity-associated differences, we analyzed ccRCC tissues from subjects with normal weight BMI 18.5–24.9 kg/m$^2$ ($n$ = 12) or WHO-defined obesity category II/III BMI $\geq$ 35 kg/m$^2$ ($n$ = 12; WHO-defined "class II/III obesity"). Doing so revealed only 36 significantly differentially expressed (DE) genes of the 521 surveyed, indicating that ~93% of genes were unaltered by obesity (Fig 4A). Of the 36 DE genes, 7 were downregulated and 29 were upregulated (Fig 4B). nano-String-generated cell type scores, which quantify relative population abundance based on gene expression, revealed no changes in T cell, CD8 T cell, NK cell, cytotoxic cell, Treg, or macrophage populations (Fig 4C). These results were largely confirmed and expanded upon at the cellular level through flow cytometric analysis of tumor-infiltrating lymphocyte/macrophage/ or myeloid (TIL/TAM/TIM) populations from a separate subset of fresh ccRCC tissues, wherein no differences were detected between ccRCC subjects with or without obesity ($n$ = 6–8 per group) (Fig 4D–4F).

## The tumor microenvironment of ccRCC subjects with obesity harbors modest, but specific alterations to the soluble protein milieu

We next surveyed the soluble protein milieu of the ccRCC the tumor microenvironment to identify any obesity-induced alterations. Of 18 proteins evaluated, two (PLGF and VEGF-A) were found to have increased concentrations within the tumor supernatant (TSN) of a subset of ccRCC subjects with obesity versus non-obese ccRCC subjects ($n$ = 17) (P = 0.0302, P = 0.0330) with the remaining 16 proteins unaltered (Fig 5A, S2 Fig). We further analyzed these proteins in the matched plasma and TSN of ccRCC subjects to determine if systemic plasma and tumor microenvironment TSN concentrations differed. Doing so revealed an increased TSN concentration of PLGF only in subjects with obesity (P = 0.0398) whereas VEGF-A was increased in TSN from both non-obese subjects (P = 0.0059) and those with obesity (P = 0.0064) subjects. These data indicate that obesity is capable of altering the soluble protein composition of the ccRCC tumor microenvironment in specific ways, but such changes are modest in frequency.

Given that *IDO1* was one of the most highly upregulated genes in our profiling of ccRCC tumors from, subjects with obesity, we also evaluated the enzymatic activity of plasma IDO to determine if IDO changes were specific to the tumor microenvironment or if these could be detected peripherally. Plasma IDO activity was not altered (P = 0.9324) in a subset of ccRCC subjects (n = 37) by obesity status, suggesting that the *IDO1* alteration is specific to the tumor microenvironment (Fig 5C). Thus, consistent with the other findings reported here, we determined that obesity has minimal effects on the cellular and soluble protein composition in either the periphery or tumor microenvironment of ccRCC subjects.

Finally, we asked whether obesity was associated with changes in the overall survival (OS) or progression-free survival (PFS) of the ccRCC subjects who were enrolled in our study. Following surgical resection of renal tumors, no statistically significant changes were found in either OS or PFS, after controlling for age, sex, and tumor stage (S3 Fig). This finding agrees

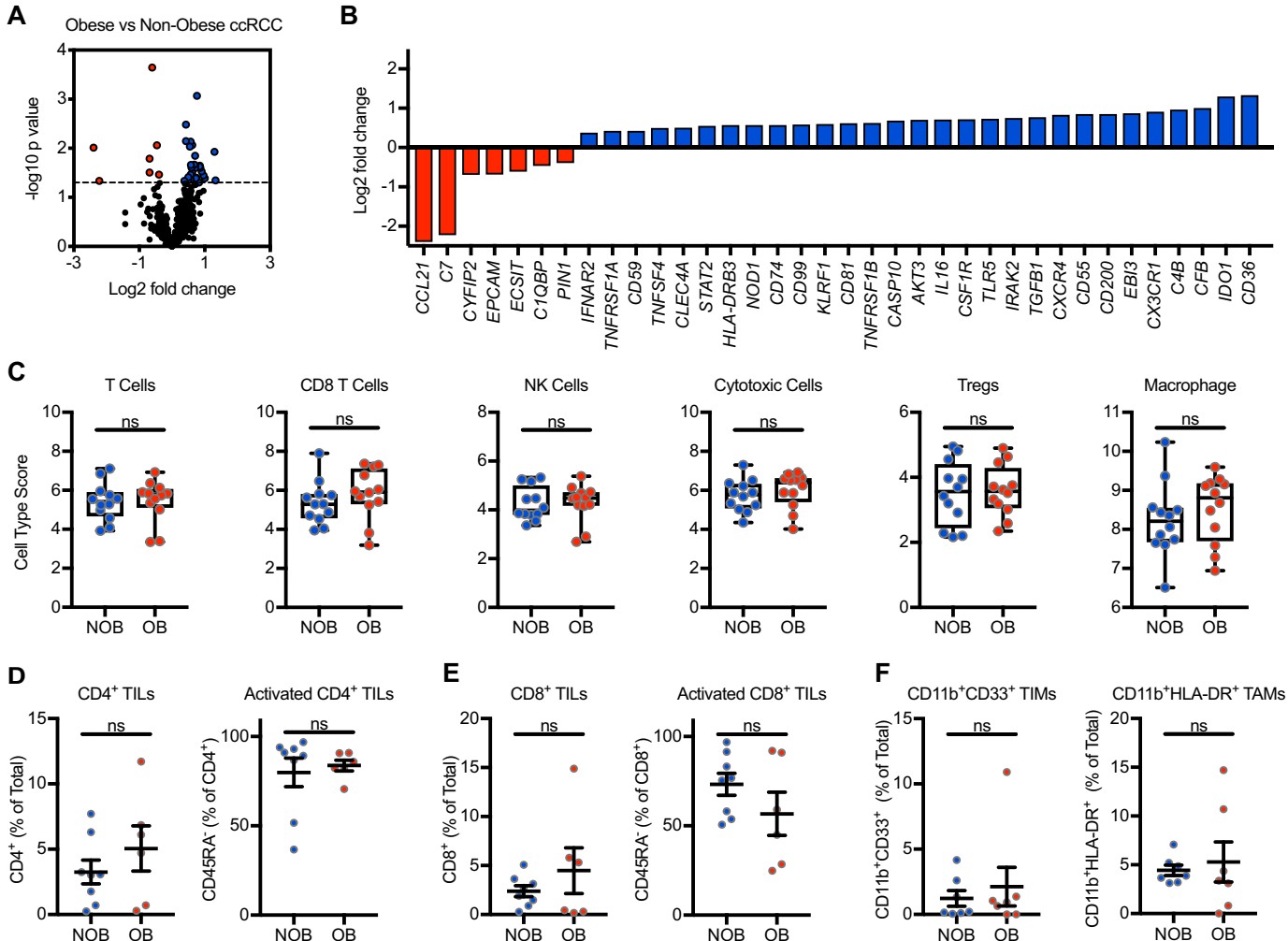

**Fig 4. Obesity induces minor alterations to the ccRCC immunogenetic profile.** (**A**) Volcano plot depicting immune-associated gene expression changes in ccRCC tissues from subjects with category II/III obesity (n = 12) versus non-obese (n = 12) subjects. Dotted line indicates p = 0.05. (**B**) Individual significantly differentially expressed genes and associated log2 fold changes from panel **A**. (**C**) nanoString-generated gene expression-derived cell type scores. Flow cytometric analysis of (**D,E**) tumor-infiltrating lymphocyte (TIL), (**F**) tumor-infiltrating myeloid (TIM) and tumor-associated macrophage (TAM) populations from ccRCC tissues of non-obese (n = 7–8) ccRCC subjects or ccRCC subjects with obesity (n = 6–7). (**C-E**) Data are presented as mean expression, boxes defining 25th to 75th percentiles with line at median and whiskers extending to minimum and maximum points, or means ± SEM with individual values for each subject shown. Statistical analyses were performed using parametric two-tailed unpaired student's t-tests or non-parametric Mann-Whitney U tests. ns = not statistically significant, NOB = Non-obese BMI<30, OB = Obese BMI≥30.

with our observation that obesity is not associated with widespread alterations in the immune profile of treatment-naive ccRCC subjects undergoing nephrectomy. Thus, neither the systemic nor tumor-infiltrating immune profiles of early-stage ccRCC patients appear to be broadly or substantially altered by obesity; those obesity-related changes we identified were limited in number and specific in scope.

## Discussion

The most recent report on obesity prevalence in the U.S. indicates that approximately 39% of adults have obesity [7]. Despite the magnitude of this healthcare problem, relatively little is known about how obesity impacts anti-tumor immunity in cancer patients. This gap in knowledge is particularly important for obesity-associated cancers, such as RCC, because their

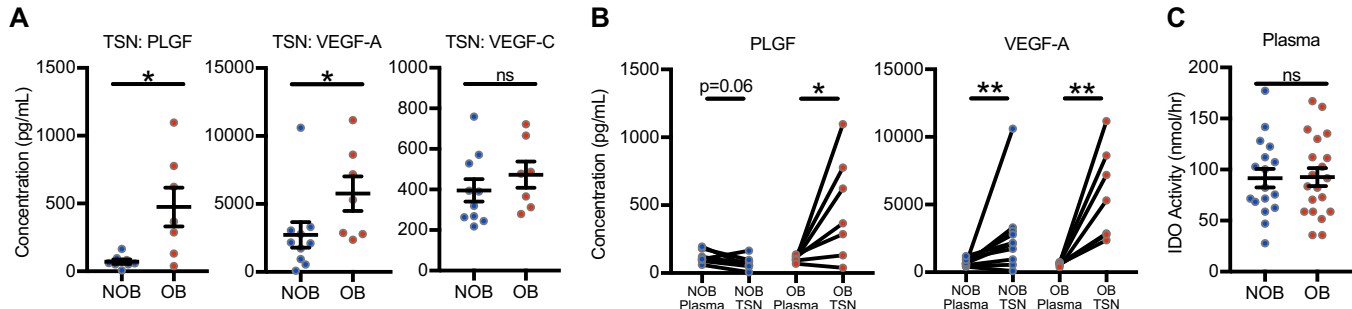

**Fig 5. Obesity alters soluble proteins specifically in the ccRCC tumor microenvironment. (A)** Unmatched analysis of protein concentrations in the tumor supernatant (TSN) from non-obese ccRCC subjects or subjects with obesity (n = 15–17). (**B**) Matched analysis of protein concentrations in plasma versus tumor supernatant (TSN) from ccRCC subjects. (**C**) IDO enzymatic activity levels in the plasma of ccRCC subjects (n = 37). Data are presented as individual values or means ± SEM with individual values for each subject shown. Statistical analyses were performed using parametric two-tailed unpaired or paired student's t-tests. ns = not statistically significant, NOB = Non-obese BMI<30, OB = Obese BMI≥30.

incidence is predicted to increase as the numbers of adults with obesity continues to rise [6, 31]. Because CPIs are FDA-approved for the management of advanced RCC, there is an urgent need to identify and understand obesity-associated alterations in immune function in these patients. Here, we investigated the effects of host obesity on the immune profile in treatment-naive ccRCC subjects with localized renal tumors. In doing so, we found that a majority of the cell types, soluble mediators, and immune-related intratumoral gene expression patterns we examined in ccRCC subjects were unaltered by obesity. Thus, obesity has only modest effects on the immune landscape in treatment-naive ccRCC subjects, a finding that suggests the smoldering, chronic inflammation that typically accompanies obesity does not result in widespread detrimental changes to either systemic or intrarenal immune parameters in ccRCC.

Of the obesity-associated changes that were present, alterations occurred in both systemic and intratumoral immune parameters in our ccRCC cohort. Systemically, we found decreased percentages of CD14+CD16neg classical monocytes and Foxp3+ regulatory T cells in ccRCC subjects with obesity (**Fig 1**). Our finding that CD14+CD16neg classical monocytes showed a negative association with obesity is of interest, due to a prior report by Krieg et al., who found that in stage IV melanoma patients, increased frequencies of circulating CD14+CD16negH-LA-DRhigh monocytes prior to anti-PD-1 therapy predicted improved treatment outcomes [25]. Although we characterized the immune landscape in early, rather than late-stage, renal cancer patients, a 2017 publication by Chevrier et al. reported that the immune response to renal cancer remains stable as tumors progress from localized to metastatic disease [32]. These authors performed high-dimensional cytometry of 75 ccRCC tumors, and determined that intra-tumoral T cell and myeloid cell populations were consistent across the spectrum of Grade I through metastatic lesions. Thus, the decrease in circulating CD14+CD16neg monocytes we found in OB subjects with localized RCC may reflect changes that are present in advanced RCC. This possibility should be examined in future studies, as should potential associations between the frequency of pre-CPI circulating CD14+CD16negHLA-DRhigh monocytes and treatment outcomes in RCC.

We also found increased concentrations of CCL3, IL-1β, and TNFα from among the 38 cytokines, chemokines, and growth factors examined in plasma from ccRCC subjects with obesity versus tumor-free donors with obesity (**Fig 2**). CCL3 is a macrophage chemoattractant [33] that has been shown to promote metastasis in a pre-clinical model of experimental renal cancer metastasis [34]. IL-1β is clinically relevant in RCC, because high intratumoral IL-1β in human renal tumors has been associated with increased MDSC accumulation [35] and poor

survival (thehumanproteinatlas.org, p <0.001). TNFα has been linked to RCC metastasis and poor outcomes to sunitinib [36], providing the foundation for clinical trials investigating TNFα blockade in RCC patients [37]. Thus, increased expression of these factors with obesity is consistent with the presence of a pro-tumorigenic systemic environment in ccRCC patients. Notably, we found that IGFBP-1 was the only factor to be decreased in the plasma of ccRCC subjects with obesity versus their non-obese ccRCC counterparts. IGFBP-1 is one of six IGF binding proteins that bind to and stabilize IGF proteins in circulation; they can have both positive and negative effects on IGF function, but IGFBP-1, specifically, has been reported to inhibit IGF-1 promotion of cancer progression [38, 39]. Thus, decreased IGFBP-1 in OB ccRCC subjects likely represents a detrimental change, as low expression of this protein may promote the pro-tumorigenic properties of IGF-1. Collectively, these obesity-associated changes in plasma proteins are consistent with the development of a mildly pro-tumor systemic environment.

To examine potential immunogenetic changes within renal tumors that may associate with obesity, we performed a nanoString-based survey of immune-related genes. To facilitate the identification of genes of interest that could be explored in future studies, we analyzed tumors from individuals who either had WHO-defined Class II/II obesity (BMI $\geq$ 35kg/m$^2$) or who were of normal weight (BMI $<$ 25kg/m$^2$), and used unadjusted P values during this screen. Of those changes identified in the tumor microenvironments of ccRCC subjects with obesity versus non-obese counterparts, those with the largest fold changes were *CD36* (+2.514, P = 0.045), *IDO1* (+2.46, P = 0.012), *CFB* (+2.0, P = 0.041), *CD7* (-4.66, P = 0.046), and *CCL21* (-5.28, P = 0.0097). CD36 is a scavenger receptor that allows cells to take up triglycerides and use them to support fatty acid oxidation, and its expression has been shown to support tumor-promoting M2 macrophages [40]. In ccRCC tumors, CD36 expression was recently found to increase as visceral fat depots increased, and high intratumoral CD36 mRNA expression was associated with worse overall survival and progression-free survival [41]. *IDO1* encodes the well-known immunosuppressive protein indoleamine 2,3 dioxygenase-1, which is currently being explored as a therapeutic target in multiple cancer types [42]. CD7 is a co-stimulatory protein receptor expressed on T cells and its expression varies with T cell activation state [43]. CCL21 (also known as Secondary Lymphoid-tissue chemokine or "SLC") is a chemoattractant for activated T cells and one study linked low expression of this chemokine to decreased OS and PFS in patients with metastatic RCC [44]. Thus, our nanoString screen suggests that in renal tumors from subjects with obesity, increases in CD36 and IDO1, with concomitant decreases in T cell-related CD7 and CCL21, are consistent with a host environment that favors tumor progression.

Despite the above potential changes in gene expression in renal tumors from subjects with obesity, it is important to note that the vast majority (93%) of examined genes, gene expression-based cell type scores, and cellular leukocytic infiltrates were found to be unaltered. Thus, our findings largely concur with those of Sanchz *et al.*, who reported that obesity did not alter the immune infiltrate in tumors from treatment-naive RCC subjects [21]. In that study, the authors performed a full transcriptomic analysis on renal tumor biospecimens, as well as immunohistochemistry and flow cytometry. Those authors found that with obesity, PD-L1 expression decreased in renal tumors, but the frequencies of CD4$^+$ or CD8$^+$ T cells, B cells, and regulatory T cells were unaltered [21]. However, in that study, the authors did not examine plasma proteins as we did here. Our findings that most plasma protein alterations were driven by the presence of a renal tumor, rather than by obesity or the combination of obesity plus a renal tumor, are consistent with our prior findings in a cohort of sarcoma subjects [16]. Thus, multiple studies support a scenario in which host obesity has a minimal impact on the systemic or intratumoral immune signature in treatment-naive individuals. Given the rising use of CPI

agents for the treatment of RCC and other tumor types, it will be critical to determine whether and how obesity alters therapy-induced immune responses in patients with advanced disease.

Associations between increasing adiposity and treatment outcomes in RCC and other cancer patients have been examined by others. The study by Sanchez *et al*. also reported on the effect of obesity on CPI outcomes in RCC patients; these authors found that after controlling for IMDC risk score, obesity as defined by BMI was not associated with alterations in CPI outcomes [21]. Our data concur with this finding, as we report here that multiple key features of the immune system (soluble proteins, cell types, gene expression patterns) were unaltered by subject obesity status, despite the fact that most of the obesity-associated changes that did occur appeared to be of a tumor-promoting nature. In contrast, a recent analysis of six independent cohorts of metastatic melanoma subjects by McQuade *et al*. concluded that obesity in men, but not women, was associated with improved overall survival or progression-free survival in response to ipilimumab plus dacarbazine; or pembrolizumab, nivolumab, or atezolizumab when used as single agents [17]. Clearly, additional work will be needed to more fully understand the implications of host obesity on anti-tumor immunity and immunotherapy outcomes in RCC and beyond.

## Conclusions

Our study has provided what we believe to be the first in-depth evaluation of the effects of obesity on systemic and tumor immune profiles of ccRCC patients. Nevertheless, our study has limitations, including the relatively small cohort size, limited probe availability precluding in-depth or high-throughput flow cytometric analysis, lack of spatial and interactive evaluation of tumor-infiltrating leukocyte populations, the restriction of our gene expression analysis to only immune-related genes, the fact that all subjects were consented from one institution in a state with a predominantly Caucasian-American demographic composition, and the relatively limited number of female subjects, which precluded our ability to determine if biological sex leads to variations in the immune parameters we examined here. In conclusion, we report that multiple immune-based analyses suggest that obesity triggers a limited number of specific, rather than widespread, pro-tumorigenic alterations to the immune landscape in treatment-naive ccRCC subjects. As such, our findings provide an important step toward understanding the complex interplay between obesity, anti-tumor immunity, and renal cancer outcomes.

## Supporting information

**S1 Fig. Elevations in plasma proteins are largely caused by the presence of ccRCC and not obesity.** Various soluble proteins were surveyed from platelet poor plasma obtained from ccRCC (n = 15–26) and tumor-free (n = 7–16) subjects. Remaining (**A**) 3/4 proteins elevated in non-obese ccRCC versus non-obese tumor-free subjects, (**B**) 5/6 proteins elevated in ccRCC subjects with obesity versus tumor-free subjects with obesity, (**C**) 11/12 proteins elevated in non-obese ccRCC subjects and ccRCC subjects with obesity versus respective non-obese and obese tumor-free controls from **Fig 3**. Data from the 15 unaltered proteins are not included. Data are presented as means ± SEM with individual values for each subject shown. Statistical analyses were performed using parametric one-way ANOVA test or non-parametric Kruskal-Wallis test followed by Bonferroni's or Dunn's multiple comparisons tests, respectively. $^*p<0.05$, $^{**}p<0.01$, ns = not statistically significant. Non-obese BMI<30, Obese BMI≥30.
(EPS)

**S2 Fig. Obesity status does not impact certain soluble proteins in the ccRCC tumor micro-environment.** Unmatched analysis of protein concentrations in the tumor supernatant (TSN) from non-obese (n = 10) and obese (n = 7) ccRCC subjects. Data are presented as means ± SEM with individual values for each subject shown. Statistical analyses were performed using parametric two-tailed unpaired student's t-tests or non-parametric Mann-Whitney U tests. ns = not statistically significant, NOB = Non-obese BMI<30, OB = Obese BMI≥30.
(EPS)

**S3 Fig. Obesity status does not alter outcomes in ccRCC patients following resection of renal tumors.** (A) Overall survival (OS) (all-cause mortality) and (B) progression-free survival (PFS) for 62 of 69 ccRCC subjects with adequate follow-up data after nephrectomy. Survival curves between ccRCC subjects without obesity (BMI < 30, blue) and with obesity (BMI ≥ 30, red) was compared by Kaplan-Meier analyses and logrank tests.
(EPS)

## Acknowledgments

The authors would like to thank the clinical research support staff in The University of Iowa Department of Urology and in The University of Iowa Molecular Epidemiology Resource Core for their assistance with this project.

## Author Contributions

**Conceptualization:** Lyse A. Norian.

**Data curation:** Justin T. Gibson, Katlyn E. Norris, Gal Wald, Claire M. Buchta Rosean, Lyse A. Norian.

**Formal analysis:** Justin T. Gibson, Katlyn E. Norris, Gal Wald, Claire M. Buchta Rosean, Lewis J. Thomas, Shannon K. Boi, Peng Li, Lyse A. Norian.

**Funding acquisition:** Lyse A. Norian.

**Investigation:** Justin T. Gibson, Katlyn E. Norris, Gal Wald, Claire M. Buchta Rosean, Lewis J. Thomas, Shannon K. Boi, Laura A. Bertrand, Megan Bing, Jessy Deshane, Lyse A. Norian.

**Methodology:** Peng Li.

**Resources:** Jennifer B. Gordetsky, James A. Brown, Kenneth G. Nepple.

**Software:** Peng Li.

**Supervision:** Lyse A. Norian.

**Visualization:** Justin T. Gibson, Peng Li.

**Writing – original draft:** Justin T. Gibson, Lyse A. Norian.

**Writing – review & editing:** Justin T. Gibson, Laura A. Bertrand, Megan Bing, Jennifer B. Gordetsky, Jessy Deshane, James A. Brown, Kenneth G. Nepple, Lyse A. Norian.

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
