## [Decision Letter · Decision Letter 0]

10 Apr 2020

PONE-D-20-05133

Obesity induces limited changes to systemic and local immune profiles in treatment-naive human clear cell renal cell carcinoma

PLOS ONE

Dear Mr. Gibson,

Thank you for submitting your manuscript to PLOS ONE. After careful consideration, we feel that it has merit but does not fully meet PLOS ONE’s publication criteria as it currently stands. Therefore, we invite you to submit a revised version of the manuscript that addresses the points raised during the review process.

A common issue raised by both reviewers concerned somewhat limited insight into the broader immune profile within tumors and functional TIL analyses provided. Therefore, an earnest responses to these comments in particular is important as you prepare your revised manuscript.

We would appreciate receiving your revised manuscript by May 9, 2020. To enhance the reproducibility of your results, we recommend that if applicable you deposit your laboratory protocols in protocols.io, where a protocol can be assigned its own identifier (DOI) such that it can be cited independently in the future. For instructions see: http://journals.plos.org/plosone/s/submission-guidelines#loc-laboratory-protocols

We look forward to receiving your revised manuscript.

Kind regards,

Ryan M. Teague, Ph.D.

Academic Editor

PLOS ONE

Journal Requirements:

2. Please provide additional details regarding participant consent. In the ethics statement in the Methods and online submission information, please ensure that you have specified what type of consent you obtained (for instance, written or verbal).

**Comments to the Author**

1. Is the manuscript technically sound, and do the data support the conclusions?

Reviewer #1: Yes

Reviewer #2: Yes

2. Has the statistical analysis been performed appropriately and rigorously? 

Reviewer #1: Yes

Reviewer #2: Yes

3. Have the authors made all data underlying the findings in their manuscript fully available?

Reviewer #1: Yes

Reviewer #2: Yes

4. Is the manuscript presented in an intelligible fashion and written in standard English?

Reviewer #1: Yes

Reviewer #2: Yes

5. Review Comments to the Author

Reviewer #1: This is a clearly written report that describes results of PBMC flow cytometric profiling and tumor genetic and flow cytometric profiling in obese vs non-obese patients with untreated ccRCC.

Among the positive features of the study are

1. The question of why and how obesity renders patients more susceptible to cancers, including RCC, is of great interest, and despite several studies of this topic, there remains no consensus with regard to the explanation or the impact of obesity on either the spontaneous immune response of patients to their tumors or the response to immunotherapy.

2. The patient and control cohorts are well defined, the experimental and analytical methods used are straightforward and the results are clearly described.

Among the limitations of the study are

1. All of the patients in the study have early stage disease and all of the studies were done at a single point in time (ie, at or shortly after diagnosis). Therefore, at this point it is impossible to know whether the observed differences between obese and nonobese patients will correlate with response to treatment or clinical course.

2. The flow based assays evaluated a relatively small number of markers among PBMC and immune cells in tumors. More advanced staining methods are available that can address many more markers, and it is possible that a more thorough analysis would have revealed additional important differences.

3. Recent studies of other cancers in which cutting edge methods have been used to analyze the tumor microenvironment have shown that the location and spatial relationships between immune cells in tumors may be more important than the relative frequencies of these cells in the tumors as a whole.

Despite these limitations, the results are of interest and may help to inform new studies in the future.

Reviewer #2: This is a well-written manuscript by Gibson et al addressing the effect of obesity on the immune profile of Renal Cell Carcinoma (RCC) patients using PBMC and tumor tissue from 69 RCC patient samples as well as healthy tissue from 38 tumor-free donors. The study has high translational significance given the current obesity epidemic and provides new insight into the obesity paradox paradigm for patients with RCC. Interestingly, the authors reached the conclusion that the systemic and intratumoral immune profile of RCC patients is independent of their BMI and hypothesize that difference may have beeen observed after immunotherapy but this was not evaluated here. The study is for the most part well designed however there are some issues that could help clarify the data and should be addressed by the authors for an overall improved manuscript:

1.  The authors reported an increase in the percentage of monocytes. Did they also look at macrophages? Was there any difference in lean vs. obese vs. healthy?

2.  In their discussion they mentioned CD36 expression increased in tumor of obese patients. Did they look at CD36 expression on any immune cell populations or was this unique to the tumor?

3.   Do the authors have any additional data on CD8 T cell function like gzmb/perforin production in both tumor and PBMC?

6. PLOS authors have the option to publish the peer review history of their article (what does this mean?). If published, this will include your full peer review and any attached files.

Reviewer #1: No

Reviewer #2: No

---

## [Author Response · Author response to Decision Letter 0]

8 May 2020

Editorial Comments:

We have reviewed the style requirements and edited the manuscript accordingly. 

2. Please provide additional details regarding participant consent. In the ethics statement in the Methods and online submission information, please ensure that you have specified what type of consent you obtained (for instance, written or verbal).

The methods section has been edited to indicate that written informed consent was obtained from all study participants. 

Statement of “data not shown” has been removed and replaced with reference to a newly added supplemental figure (S2 Fig) which contains the referenced data that was previously not shown. 

Reviewer #1: This is a clearly written report that describes results of PBMC flow cytometric profiling and tumor genetic and flow cytometric profiling in obese vs non-obese patients with untreated ccRCC.

Among the positive features of the study are

1. The question of why and how obesity renders patients more susceptible to cancers, including RCC, is of great interest, and despite several studies of this topic, there remains no consensus with regard to the explanation or the impact of obesity on either the spontaneous immune response of patients to their tumors or the response to immunotherapy. 

We thank the reviewer for this positive comment and strongly agree that this is an important area of research that requires further investigation. We believe our manuscript contributes much-needed insight into this topic.

2. The patient and control cohorts are well defined, the experimental and analytical methods used are straightforward and the results are clearly described. 

Thank you. We are delighted to hear these positive comments regarding our initial submission.

Among the limitations of the study are

1. All of the patients in the study have early stage disease and all of the studies were done at a single point in time (ie, at or shortly after diagnosis). Therefore, at this point it is impossible to know whether the observed differences between obese and nonobese patients will correlate with response to treatment or clinical course.

The reviewer points out an important detail in the design of our study. It is true that our biospecimens were obtained from study subjects with early stage tumors at the time of nephrectomy. Unfortunately, in this study, it is not possible for us to obtain renal tumor specimens from patients who have been diagnosed with late-stage/disseminated disease, as it goes against standard practice to perform renal biopsies on these patients. 

Our attempts to determine the impact of obesity on clinical outcomes in this small cohort of patients provided no clear indication of either beneficial or detrimental responses. Of 69 subjects evaluated, 62 had sufficient follow-up data, with the vast majority (�80%) reaching overall survival (all-cause mortality) of 5 years. We found no significant differences in either overall survival or progression-free survival in RCC patients with obesity relative to individuals without obesity. This is not surprising given the limited number of patients being evaluated. Notably, these non-significant results do support our overall finding that obesity is associated with a limited number of changes in the immune profiles of treatment-naive RCC patients with early-stage disease. These data have been included as a new supplemental Figure 3.

2. The flow based assays evaluated a relatively small number of markers among PBMC and immune cells in tumors. More advanced staining methods are available that can address many more markers, and it is possible that a more thorough analysis would have revealed additional important differences. 

We concur that more advanced flow cytometric staining methods are available. However, during the time at which samples for this observational study were obtained and analyzed, we were limited to performing only standard, multi-parameter flow cytometry. Please note that our analyses are equally or more detailed than those provided in the supplementary data of the 2020 Sanchez et al. paper published in The Lancet Oncology. To address this concern, we have modified the Discussion (lines 529-530 of the revised manuscript) to address this limitation in our study. 

3. Recent studies of other cancers in which cutting edge methods have been used to analyze the tumor microenvironment have shown that the location and spatial relationships between immune cells in tumors may be more important than the relative frequencies of these cells in the tumors as a whole.

We agree with the reviewer that examining the spatial relationships among tumor-infiltrating leukocytes would very likely generate interesting results. Unfortunately, a limited amount of tumor tissue was available for analysis in our study and we chose to use it for performing nanoString immunogenetic profiling, multiparameter flow cytometric analysis on TILs, and multiplex analyses on supernatants from homogenized tumors. However, from a limited number of tumor biospecimens we were able to conduct preliminary analyses using immunofluorescent staining to look at spatial distribution of leukocyte populations. No obvious differences were noted. However, due to the small sample size, we feel the data are too preliminary to use for drawing any meaningful clinical conclusions at present. Given the time constraints on resubmission, we were not able to obtain additional tumor tissues to address this point. We have modified the manuscript (lines 530-531 of the revised manuscript) to call attention to this limitation. 

Reviewer #2: This is a well-written manuscript by Gibson et al addressing the effect of obesity on the immune profile of Renal Cell Carcinoma (RCC) patients using PBMC and tumor tissue from 69 RCC patient samples as well as healthy tissue from 38 tumor-free donors. The study has high translational significance given the current obesity epidemic and provides new insight into the obesity paradox paradigm for patients with RCC. Interestingly, the authors reached the conclusion that the systemic and intratumoral immune profile of RCC patients is independent of their BMI and hypothesize that difference may have been observed after immunotherapy but this was not evaluated here. The study is for the most part well designed however there are some issues that could help clarify the data and should be addressed by the authors for an overall improved manuscript:

We thank the reviewer for these positive comments and suggestions for improving our manuscript. 

1. The authors reported an increase in the percentage of monocytes. Did they also look at macrophages? Was there any difference in lean vs. obese vs. healthy?

To clarify, the percentages of peripheral blood classical (CD14+CD16-) monocytes were found to be decreased in obese vs non-obese ccRCC subjects and obese ccRCC vs obese tumor-free subjects, but were unaltered by obesity in tumor-free donors (Fig. 1F). In contrast, the percentages of circulating non-classical (CD14-CD16+) monocytes were found to be unaltered by obesity in RCC subjects (Fig. 1C).

In terms of performing an analysis of macrophages, we are limited to our preexisting analysis of renal tumor tissues. As such, we are unable to examine changes in RCC patients vs healthy controls, as inquired. We have included new data in Figure 4F, showing that obesity does not alter the percentages of CD11b+HLA-DR+ tumor associated macrophages (TAMs) in renal tumors. These data agree with our nanoString generated cell type scores which revealed no difference in the relative abundance of macrophages in non-obese vs obese ccRCC subjects (Fig. 4C). 

2. In their discussion they mentioned CD36 expression increased in tumor of obese patients. Did they look at CD36 expression on any immune cell populations or was this unique to the tumor? 

This is a very interesting question that we are unfortunately unable to answer at present. We did not include CD36 in our analysis of leukocyte populations in circulation or within renal tumors, as we prioritized major leukocyte population markers (e.g. CD4, CD8, CD11b). Thus, our information on this marker is limited to our nanoString data at present. We agree that future examination of CD36 expression patterns on tumor-infiltrating leukocyte populations would be worthwhile, given our nanoString findings. Sadly, as we used our fresh tumor tissue for other analyses, we have no additional tissue available for use in answering this question. 

3. Do the authors have any additional data on CD8 T cell function like gzmb/perforin production in both tumor and PBMC?

Our current functional analysis of CD8 T cells is limited to that presented in Fig. 2 in regard to PBMCs. However, comparing the functional cytotoxic capacity of peripheral blood and intratumoral CD8 T cells is intended for future follow up studies.

---

## [Decision Letter · Decision Letter 1]

13 May 2020

Obesity induces limited changes to systemic and local immune profiles in treatment-naive human clear cell renal cell carcinoma

PONE-D-20-05133R1

Dear Dr. Gibson,

We are pleased to inform you that your manuscript has been judged scientifically suitable for publication and will be formally accepted for publication once it complies with all outstanding technical requirements.

With kind regards,

Ryan M. Teague, Ph.D.

Academic Editor

PLOS ONE

Additional Editor Comments (optional):

Reviewers' comments:

Reviewer's Responses to Questions

**Comments to the Author**

1. If the authors have adequately addressed your comments raised in a previous round of review and you feel that this manuscript is now acceptable for publication, you may indicate that here to bypass the “Comments to the Author” section, enter your conflict of interest statement in the “Confidential to Editor” section, and submit your "Accept" recommendation.

Reviewer #2: All comments have been addressed

2. Is the manuscript technically sound, and do the data support the conclusions?

Reviewer #2: Yes

3. Has the statistical analysis been performed appropriately and rigorously? 

Reviewer #2: Yes

4. Have the authors made all data underlying the findings in their manuscript fully available?

Reviewer #2: Yes

5. Is the manuscript presented in an intelligible fashion and written in standard English?

Reviewer #2: Yes

6. Review Comments to the Author

Reviewer #2: (No Response)

7. PLOS authors have the option to publish the peer review history of their article (what does this mean?). If published, this will include your full peer review and any attached files.

Reviewer #2: No

---

## [Editor Report · Acceptance letter]

14 May 2020

PONE-D-20-05133R1 

Obesity induces limited changes to systemic and local immune profiles in treatment-naive human clear cell renal cell carcinoma 

Dear Dr. Gibson:

I am pleased to inform you that your manuscript has been deemed suitable for publication in PLOS ONE. Congratulations! Your manuscript is now with our production department. 

With kind regards,

on behalf of

Dr. Ryan M. Teague 

Academic Editor

PLOS ONE